# Efficacy of Intensive Inpatient Therapy in Infants with Congenital Muscular Torticollis Involving the Entire Sternocleidomastoid Muscle

**DOI:** 10.3390/children10061088

**Published:** 2023-06-20

**Authors:** Dong Rak Kwon, Sung Cheol Cho

**Affiliations:** Department of Rehabilitation Medicine, Catholic University of Daegu School of Medicine, Daegu 42472, Republic of Korea

**Keywords:** congenital muscular torticollis, intensive treatment, cervical passive range of motion, ultrasound

## Abstract

The efficacy and frequency of physiotherapy in the prognosis of congenital muscular torticollis (CMT) that involves the entire sternocleidomastoid (SCM) muscle continues to be unclear. This study investigated the therapeutic effect of intensive inpatient therapy given to infants with CMT that involves the whole SCM using clinical measurements and ultrasound (US). This study included 54 infants (27 boys and 27 girls; mean corrected age of 18.57 days) evaluated for CMT at our outpatient clinic from January 2014 to May 2021. The included patients were divided into three groups (groups 1, 2, and 3). Patients in group 1 underwent outpatient treatment 12 times. Patients in groups 2 and 3 underwent therapeutic exercise followed by US diathermy with microcurrent twice daily for 1 or 2 weeks, respectively. Passive range of motion of the cervical rotation (PCRROM) and SCM thickness were evaluated pre- and post-treatment. Among the three groups, the demographic data at baseline were not significantly different, SCM thickness and PCRROM were significantly decreased/increased at post-treatment compared to pre-treatment (*p* < 0.05), mean PCRROM change was significantly greater in group 3 (*p* < 0.05), and mean SCM thickness reduction between pre-treatment and 3 months post-treatment was significantly greater in groups 2 and 3 (*p* < 0.05). Therefore, intensive inpatient therapeutic exercise and US diathermy with microcurrent may enhance the prognosis of CMT involving the entire SCM muscle.

## 1. Introduction

Congenital muscular torticollis (CMT) is the third most common congenital anomaly among infants, only after congenital hip dysplasia and clubfoot. The reported incidence of CMT varies between 0.4% and 2.0% [1]. This condition involves the neck and arises from a shortened or excessively contracted sternocleidomastoid muscle (SCM) [1,2,3]. The compromised length of the SCM, either with or without a palpable mass, can happen due to muscle trauma during birth or prolonged poor intrauterine posture, which leads to chronic repetitive microtrauma [4,5]. Any injuries at the time of delivery can result in SCM tear, accompanied by hematoma formation and subsequently leading to fibrous contracture [6]. In contrast, restricted head mobility during intrauterine development can give rise to progressive neck contracture, which can subsequently lead to the development of muscle fibrosis [7]. There is a brief alteration in calcium concentrations following muscle trauma that plays a vital role in muscle excitation–contraction coupling. Prolonged elevation of calcium concentration within the muscles can result or enhance the structural changes, including excessive fibrosis in the endomysium and perimysium, as well as muscle fibre atrophy. Consequently, this fibrosis causes tightening of the affected SCM and restricts the cervical active and passive range of motion [8]. The extent of muscle fibrosis is closely linked to the age of the patient when left untreated. With the passage of time, the presence of fibrosis becomes more prominent [8]. Furthermore, it was observed that age is a contributing factor or determinant for pathogenesis and features like fibrosis and adipogenesis, among patients with CMT. As a result, early diagnosis and treatment of CMT lead to positive clinical outcomes, and regular physiotherapy treatment can help restore both the length and SCM functional status [8,9].

Recent reports have drawn attention to secondary changes occurring in the cervical spine of individuals with CMT. Previous studies have indicated that deformities in the cervical spine and craniofacial region can emerge as early as 8 months of age. Furthermore, the severity of these deformities intensifies with age and correlates with the degree of tightness in the SCM muscle [10].

In case of fibrosis of the entire SCM, it is estimated that approximately one-third of patients cannot be effectively treated with physical therapy alone. Consequently, more invasive interventions like surgery and botulinum neurotoxin injections may be necessary to treat the contracture [11]. Active or passive stretching of the affected muscle may only yield positive results in infants with involvement of less than two-thirds of the SCM muscle [11]. Therefore, a novel conservative treatment approach is imperative to reduce the need for surgery and invasive procedures in infants with CMT displaying fibrosis involving the whole SCM.

A previous study demonstrated that a 100-time daily stretching regimen resulted in greater improvements in head position and PCRROM compared to a 50-time stretching routine among infants with CMT [12]. Moreover, our previous investigation revealed that combining stretching exercises with ultrasound (US) and microcurrent therapy (MCT) in infants with CMT yielded superior outcomes in terms of neck range of motion and therapeutic compliance compared to stretching exercises combined with US alone [13]. However, no prior studies have compared the effectiveness of intensive inpatient treatment durations in infants with CMT exhibiting fibrosis involving the entire SCM.

These findings emphasize the importance of developing alternative treatment strategies that can effectively address the challenges posed by CMT with fibrosis affecting the entire SCM muscle. By further exploring the potential benefits of intensive inpatient treatment durations, researchers may uncover valuable insights into optimizing therapeutic approaches for this specific patient population.

MCT has been found to have therapeutic effects on muscle damage, attributed to various mechanisms. Firstly, it is believed that microcurrent therapy helps maintain intracellular calcium (Ca^2+^) balance following muscle damage, as demonstrated in a previous study [14]. By regulating intracellular calcium levels, this therapy can potentially prevent disruptions to the muscle membrane integrity and detrimental changes in muscle function caused by increased calcium concentration [15]. In the context of CMT, where sustained isometric contraction of the SCM can impede regular blood flow and ion exchange at the cellular membrane, this mechanism becomes particularly relevant [16,17].

Secondly, MCT has the potential to improve ATP production, facilitate amino acid transportation, and promote protein production. These processes play crucial roles in reducing inflammation and promoting tissue healing [18,19]. Previous studies have also highlighted the beneficial effects of electric current in repairing soft tissues [20].

Lastly, MCT may influence the cell nuclei activity and triggers the genes responsible for regulating collagen depletion and breakdown, thus alleviating fibrosis. Notably, in patients with head and neck carcinoma with radiation-induced fibrosis, the initiation of microcurrent therapy resulted in a significant improvement in cervical rotational range of motion [18].

Previous studies have indeed supported the efficacy of therapeutic US in reducing inflammation and enhancing blood circulation in treated tissues, if the application is conducted appropriately [21,22]. Furthermore, a recent animal study provided additional evidence of the benefits of therapeutic US. The study demonstrated that applying therapeutic US was efficient in protecting the muscle fibres from undergoing oxidative stress, mitigating the inflammatory reactions, and promoting the reorganization of muscle tissues, as observed by immunohistochemical analyses [23].

Ultrasonography (USG) has proven to be a valuable diagnostic tool due to its accuracy in detecting morphological changes and muscular pathologies, including myopathy and cerebral palsy [24,25,26,27,28]. In the context of CMT, B-mode USG of the affected SCM can provide crucial information for assessment. This includes evaluating echogenicity, texture, as well as the transverse and longitudinal extents of involvement. In comparison to the normal contralateral side muscle, the echo texture of the affected muscle is typically more hyperechoic. The degree of hyperechogenicity is a notable and consistent feature that correlates with the severity of CMT. Hyperechogenicity among the muscles of patients with neuromuscular conditions is primarily attributed to the substitution of muscle tissue with fat and fibrous tissues [28].

Our hypothesis suggests that a two-week inpatient program, characterized by intensive treatment, is expected to yield superior treatment outcomes compared to a one-week intensive treatment program or an outpatient-based treatment approach.

This study primarily aimed to assess the efficacy of intensive inpatient treatment in infants with CMT affecting the entire SCM using standard clinical outcome tools and USG. A secondary aim of this research was to compare the short-term efficacy of two different intensive inpatient treatment durations.

## 2. Methods

### 2.1. Participants

This retrospective chart review identified 1131 potential participants who visited our outpatient clinic with symptoms of torticollis from January 2014 to May 2021. The institution conducting the study was a tertiary care centre, and all patients included in the study were referred with symptoms of torticollis from other healthcare providers. Referred patients with a palpable mass on physical examination were diagnosed with torticollis involving the entire SCM muscle based on imaging findings such as USG and plain radiographs. The research was proposed to and approved by the Institutional Review Board (IRB) and the Independent Ethics Committee of the University Medical Center (IRB No. CR-22-013). Patient consent was not considered as this was a retrospective study design.

This study recruited infants with CMT from the outpatient clinic of the university hospital and who met with the following selection criteria: (1) impairment of entire SCM muscle; (2) SCM muscle thickness of >10 mm, measured through USG; (3) a palpable mass or lump in SCM which was obvious during palpatory assessment; (4) infant < 3 months of age; and (5) no history of rehabilitation care before study participation.

Patients with (1) postural torticollis, (2) congenital abnormalities of cervical spine, (3) spasmodic torticollis, (4) disorders of developmental delay like cerebral palsy and intellectual compromise, and (5) ocular torticollis were excluded from the study.

We recommended hospitalization within one month of birth. Although we recommended a standard treatment period of 2 weeks for all patients, the duration varied in some cases based on their parents’ preferences.

Of the 1131 patients with torticollis, 1077 were excluded. According to the exclusion criteria, 837 infants with postural torticollis, 29 with neurodevelopmental disorders, 3 with congenital abnormalities of the cervical spine, 2 with ocular torticollis, and 1 with spasmodic torticollis were excluded. In total, 259 patients fulfilled the selection criteria. A total of 205 patients could not wait for hospitalization and went to other hospitals, 15 received one week of intensive treatment, and 10 did not want the intensive treatment. Finally, we analysed 54 patients: 10 patients in the group who did not receive intensive care in the hospital (group 1), 29 patients in the 1-week intensive care group (group 2), and 15 patients in the 2-week intensive care group (group 3) (Figure 1).

### 2.2. Procedures

Group 1 was treated using therapeutic exercise and US and microcurrent therapy three times a week for 4 consecutive weeks (total: 12 times), consisting of US diathermy application to the affected SCM for 5 min, at 1.0 MHz and intensity of 0.8 W/cm^2^, effective radiating area of 1 cm^2^, and 50% duty cycle of 1:1 (5 ms on, 5 ms off), followed by therapeutic exercises. Each session of exercises were performed for 20 min and comprised ROM exercises, postural re-education, and mild manual stretching of the SCM. A standardized protocol was followed for the manual stretching by the physical therapist (PT) who was trained in paediatric neuromuscular rehabilitation; this protocol consisted of manual stretching with a 10 s hold period and interposed rest for 5–10 s. This was repeated for 3 sets of 15 stretches.

Each MCT (EMI; Cosmic Co., Seoul, Republic of Korea) session was provided for 30 min 3 times a week by a qualified PT [29]. The MCT generator was pre-programmed to render an alternating current with monophasic rectangular pulses at 8 Hz, polarity reversal every 3 s, and an intensity of 200 µA. This current intensity value was very much below the infants’ brink of sensation. The SCM on the treatment side was located by turning the participant’s head toward the opposite side to enhance palpation and localisation of the SCM for electrical patch attachment. The head was returned to the original position after this procedure to prevent SCM stretching effects during the MCT.

Groups 2 and 3 underwent the same rehabilitation program twice per day for 1 week (total: 10 times) or 2 weeks (total: 20 times) during the hospitalization period, respectively. After being discharged, bilateral cervical lateral bending and rotatory ROM exercises were advised to parents as a home exercise program.

During the first visit to our outpatient clinic, the physiatrist educated all parents about the home exercise program, and the PT provided further instructions on the program at the end of each treatment session. These exercises were to be repeated 10 times each session, with 6 sessions/day. Apart from this, as a part of the home exercise program, the parents were taught how to position and handle the patient, as it will lead to AROM of cervical rotation toward the affected side and induce the infants to tilt their head toward the affected side. The parents were educated to persistently change the infant’s sleeping posture between side and prone lying positions. Home exercise data were gathered through monthly exercise documentation sheets that were completed and filed by the parents. These data were retrieved and analysed by the physiatrist upon every clinic visit. Performing the exercises 6 times/session every day was considered good adherence to exercise.

All infants were scheduled for regular follow-up examinations with the physiatrist for consecutive 4 weeks. Apart from this, the infants were rescheduled for examination by the physiatrist within 7 days if the PT evaluated a normal passive CRROM over the affected side during the course of treatment. The rehabilitation treatment was discontinued if patients met the following norms: (1) difference in PCRROM is <5° of the normal side; (2) bilaterally symmetrical movement patterns were recorded; (3) normal motor development with respect to age was attained; (4) no head tilt on active or static position was seen; and (5) the parents understood what to monitor for as the child grows [30]. The home exercise program was recommended to be continued throughout the treatment period, and we strongly encouraged parents to continue the program even after the treatment was discontinued. The duration of the home exercise program was not standardized and varied among patients based on their individual needs.

### 2.3. Outcome Measurements

An arthrodial protractor was utilised to gauge the PCRROM of the affected side by a physiatrist blinded to the group allotment during the study. The infant was positioned in supine position over the examination table with shoulder stabilization. The examiner stabilised the cervical spine and head in a neutral position on the edge of the examination couch. The cervical spine could be rotated in all planes with the infant in this position.

A previous study [31] revealed a correlation coefficient of interexaminer reliability of 0.71 in measuring the passive range of motion of the neck in infants using this method. We designated ≥100° as the reference value for normal PCRROM based on an earlier study [32].

PCRROM was measured before treatment and 1 month after treatment in group 1. The PCRROM of the affected side was evaluated before and immediately after the last treatment in groups 2 and 3.

All USG examinations were conducted by a physiatrist with 20 years of experience in musculoskeletal USG using a commercially available USG system with a 5- to 18-MHz multifrequency linear transducer (EPIQ 5; Philips Healthcare, Andover, MA, USA). All ultrasonographic images were reviewed by the same physiatrist. USG was performed after the infants slept with the help of their mother. The infant was lying over the couch and the examiner observed the infant’s head posture from above. The neck was extended maximally using a small bolster, and the head was rotated from the opposite side to the examined side. USG was withdrawn if the infant was tense and did not cooperate.

Both a longitudinal scan, which was longitudinal to the affected muscle, and transverse scan, which was at a right angle to the affected muscle, were conducted on each infant. The transverse scan comprised the levels of origin at the clavicle and sternum and insertion at the mastoid process, and also the muscle belly. The site of the mass in the sternal or clavicular component of the SCM and its location in the lower, middle, or upper third of the SCM were documented.

The thickness (Figure 2), defined as the distance between the superficial and deep aponeurosis in the thickest part of the impaired SCM, was measured on a transverse scan. SCM thickness of the impaired and normal sides was analysed using US pre-treatment and 3 months post-treatment. USG scanning was performed two times, and one typical image from the scan was utilized to determine the intra-rater reliability of the SCM thickness.

### 2.4. Statistical Analysis

Data analysis and interpretation were conducted with the help of SPSS software version 19.0 (SPSS, Inc., Chicago, IL, USA) and R version 4.1.2, with the significance level set at <0.05. The central tendency and dispersion preferred was the mean and standard deviation. The intra-rater reliability of the repeated measurements of SCM thickness was assessed using the intraclass correlation coefficient. Paired *t*-test and Wilcoxon signed rank test were used to compare the parameters before and after treatment within each group. The differences between the three groups in terms of clinical and US parameters (PCRROM and SCM thickness) were evaluated using one-way analysis of variance (OW-ANOVA) and the Kruskal–Wallis test (KW-test), and an independent *t*-test and Mann–Whitney U test were employed to compare the inpatient and outpatient treatment groups. The post hoc analysis was performed using Bonferroni’s test (BFT).

## 3. Results

Group 1 had 10 infants (6 boys and 4 girls; affected side, 6 right, 4 left; mean corrected age, 19.60 ± 5.52 days), group 2 had 29 infants (13 boys and 16 girls; affected side, 15 right, 14 left; mean corrected age, 20.40 ± 9.01 days), and group 3 had 15 infants (8 boys and 7 girls; affected side, 7 right, 8 left; mean corrected age, 18.34 ± 15.22 days). There were no statistically significant differences in corrected age (*p*-value = 0.873), sex (*p*-value = 0.794), or affected side (*p*-value = 0.618) between the three groups. The demographic data at baseline among the three groups were not significantly different.

In the inpatient treatment groups (group 2 and 3), both the thickness of SCM and the PCRROM showed a significant decrease/increase when compared to the outpatient treatment group (group 1) (*p* < 0.05, Table 1).

All three groups had a significant decrease in SCM thickness following the respective group intervention (*p* < 0.05, Table 2). Additionally, all three groups had a significant increase in PCRROM after treatment compared to before treatment (*p* < 0.05, Table 3). The greatest mean change in PCRROM was observed in group 3 (16.33° ± 5.50°), followed by group 2 (9.55° ± 2.13°) and group 1 (6.00° ± 3.43°) (*p* < 0.05, Table 2). Furthermore, the mean reduction in SCM thickness between the pre- and 3 months post-treatment periods was significantly greater in group 2 (−3.18 mm ± 2.10 mm) and group 3 (−3.60 mm ± 2.21 mm) than in group 1 (−1.55 mm ± 1.28 mm) (*p* < 0.05, Table 3).

The intraclass correlation coefficient values for repeated SCM thickness measurements using US were 0.912, 0.917, and 0.928 for groups 1, 2, and 3, respectively.

## 4. Discussion

The prominent finding of this research was the significantly decreased/increased SCM thickness and PCRROM after treatment with the rehabilitation program compared with those before treatment in the three groups. An improved PCRROM was observed using intensive inpatient therapy among infants with CMT involving the entire SCM muscle twice per day for 2 weeks (group 3), followed by twice per day for 1 week (group 2) and three times per week for 4 weeks (group 1). These results could be attributed to several factors. First, the number of treatment sessions [20] in group 3 was higher than that in groups 2 [10] and 1 [12]. The literature supports maximum stretch repetitions (100 stretches/day) for a better prognosis in the head tilt and PCRROM [12]. This finding was consistent with our results. Second, the management of the home program in group 3 was better than that in groups 1 and 2. The home program can be more efficient with the help of positive feedback from physiatrists and PTs at the hospital. A successful outcome primarily depends on good cooperation with the parents [33]. Third, the microcurrent treatment included in the rehabilitation treatment program may have an effect. Our previous study [13] revealed a significantly decreased treatment duration in the same target population after the addition of MCT to the rehabilitation protocol that also predominantly included exercise and US. The average intervention time was 2.6 months in infants who were treated with MCT in addition to therapeutic exercise and US and 6.3 months among infants who were treated with therapeutic exercise and US alone. Early improvement in PCRROM was observed as early as 1 month following treatment with the addition of MCT. Therefore, we evaluated PCRROM at 1 month in group 3 and immediately after the final treatment in the other two groups.

The exact mechanism of MCT in improving muscle fibrosis is not well understood. However, many mechanisms justifying the therapeutic effect of MCT have been documented. MCT enhances the movement of extracellular calcium ions to penetrate the cell membrane, and a higher level of intracellular calcium encourages ATP synthesis. Eventually, protein synthesis is induced by mechanisms that control DNA, thus promoting cell repair. This mechanism can be applied to CMT, in which sustained contraction of the SCM negatively influences normal blood circulation and ion exchange at the cell level [18,19].

In this study, infants less than one month of age were selected, because the prognosis for CMT is generally influenced by the age of commencement of intervention [31]. CMT may not spontaneously resolve and they need comprehensive physical therapy interventions, including various exercises to the cervical and trunk musculature to encourage symmetrical movement patterns, cervical stretching and strengthening, environmental adaptation, and parent education. It has been documented that 98% of infants achieve normal cervical ROM in 1.5 months if treated with early physiotherapy intervention [9]. It has already been demonstrated that the nature of the soft tissue that replaces the degenerated muscle tissue differs depending on the patient’s age [10]. Muscle fibres are replaced by immature cellular fibrous tissue, which matures and solidifies over time. Additional MCT might enhance the rate at which the fibrotic SCM is replaced with normal muscle fibres, enhancing a structural change and thereby increasing the PCRROM.

USG has been performed as a useful imaging technique to analyse the morphology of musculature in the human body. Park et al. [34] demonstrated that the density of the SCM in B-mode USG correlated well with PROM of neck muscles. Analysing the PROM of the neck with the help of a goniometer is not difficult but sometimes it can take longer with uncooperative infants. Our study revealed a significantly lower SCM muscle thickness at 3 months post-treatment in infants who received an inpatient intensive rehabilitation program than that in the patients who underwent outpatient care. These favourable modifications in muscle properties were found to be successfully semi-quantified through USG and confirmed by the change in PCRROM.

In the group receiving inpatient treatment, we observed a positive association between the improvement in PCRROM after intensive treatment and the decrease in SCM thickness as seen on USG after 3 months. This suggests that the initial intensive treatment likely alleviated the range of motion limitations, consequently enhancing the children’s adherence to physical therapy and home exercise.

In our study, we proposed a standardized treatment duration of two weeks for all patients. It is important to note that the optimal length of inpatient treatment for this particular condition is not well established. Our observations indicate that there were difficulties in sustaining hospitalization for longer than two weeks. Consequently, the actual duration of hospitalization varied among patients, taking into consideration the preferences of parents or caregivers involved in the decision-making process.

Our study had some limitations. Since it is a retrospective study, it lacked a control group; therefore, comprehensive comparisons to other studies may be challenging. Second, the number of patients was restricted, and future validation through large-scale studies is warranted. Third, the study outcomes cannot be generalized to all subjects with CMT due to our cohort comprising only infants with CMT of the entire SCM muscle. Hence, studies should be performed on patients with congenital torticollis. Fourth, we were not able to measure the interrater reliability of the USG measurements. USG has been found to be operator-dependent, with an inherently long learning curve, and there are technical issues in image reproducibility based on the experience of the examiner. However, a physiatrist with 20 years of musculoskeletal USG experience performed all the examinations in this study. Fifth, infants are required to be immobile while performing the USG examination to avoid motion artifacts, but it is sometimes very difficult. Here, all images were extracted after the infants had fallen asleep. Sixth, home exercise data were gathered through monthly exercise recording sheets that were completed by the parents. The use of digital recording in real time via mobile applications would make the outcome more reliable. Seventh, we did not consider the influence of home exercise program duration. The efficacy of the home exercise program beyond the treatment period needs to be investigated in future studies. Eighth, we only evaluated PCRROM as the outcome measurement. Evaluation of cervical lateral flexion range of motion would have made the result more reliable.

Ninth, further research is necessary to assess the effects of MCT using different current intensities, waveforms, frequencies, and treatment durations. This research will help identify the optimal parameters that yield the most effective and consistent results. Tenth, we did not perform sample size calculations because there was no previous study like our study. We did not perform retrospective power analysis. The power for cervical ROM was 1.0. But the power for SCM thickness was 0.64. This means a larger sample size is necessary for exact results. In future studies, we will recruit a larger infant study population. Finally, infants whose treatment started after 1 month of age were not included in this study. Generally, the prognosis for CMT is dependent on the age at which the rehabilitation program was started. Further research is necessary to analyse the effects of rehabilitation programs according to the age at which the treatment is initiated in infants with CMT of the entire SCM muscle.

In conclusion, to the best of our knowledge, this is the first research that correlates the short-term impacts of two intensive inpatient rehabilitation treatments in infants with CMT of the entire SCM muscle. The current study demonstrated that intensive inpatient treatment may improve the efficiency of therapeutic exercise and US with microcurrent therapy for the treatment of CMT of the entire SCM muscle.

## Figures and Tables

**Figure 1 children-10-01088-f001:**
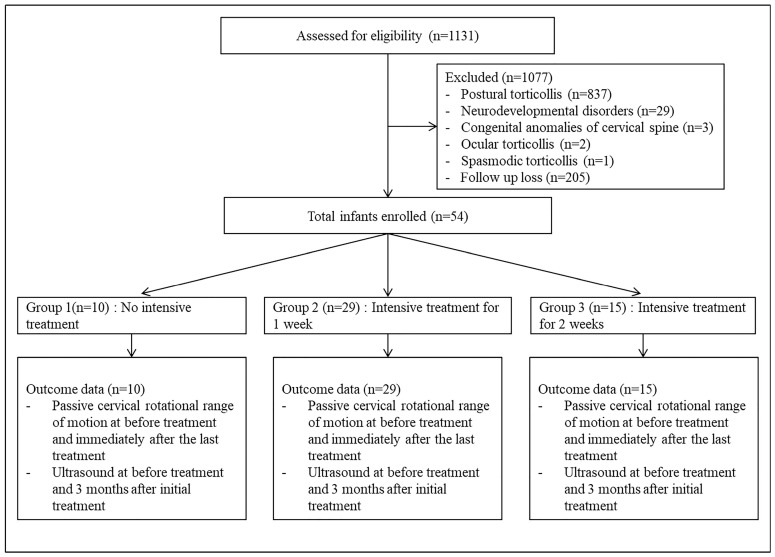
Flow chart of the study.

**Figure 2 children-10-01088-f002:**
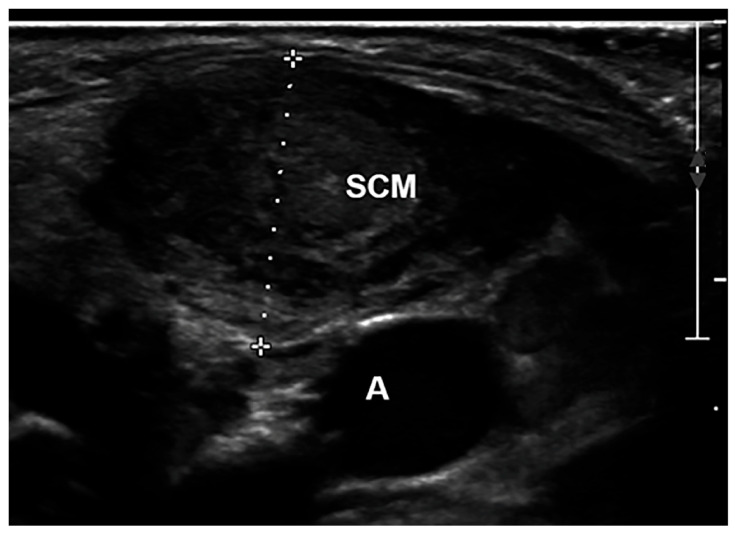
Ultrasound of the SCM. The thickness is the distance between the superficial and the deep aponeurosis of SCM. A indicates the carotid artery.

**Table 1 children-10-01088-t001:** Comparison of changes in the passive cervical rotational range (°) and SCM thickness between outpatient and inpatient treatment groups.

	Outpatient Treatment (*n* = 10)		Inpatient Treatment (*n* = 44)	
	Mean	SD	Mean	SD
Cervical rotational PROM (°)				
Pre-treatment	63.50	10.53	74.89	5.86
Immediately after the last treatment	69.50	9.07	86.59	3.70
Δ	6.00	3.43	11.70 *	4.94
SCM thickness (mm)				
Pre-treatment	13.36	2.36	11.97	1.48
3 months post-treatment	11.81	2.60	9.44	2.07
Δ	−1.55	1.28	−3.58 *	1.85

Outpatient treatment, patients who did not undergo intensive treatment; inpatient treatment, patients who undergo intensive treatment. Δ, difference between pre-treatment and post-treatment. SD, standard deviation. * *p* < 0.05 according to independent *t* test and Mann–Whitney U test between inpatient and outpatient treatment patients.

**Table 2 children-10-01088-t002:** Comparison of changes in the passive cervical rotational range (°) among the three groups.

	Group 1 (*n* = 10)		Group 2 (*n* = 29)		Group 3 (*n* = 15)	
	Mean	SD	Mean	SD	Mean	SD
Before treatment	63.50	10.53	76.21	5.11	72.33	6.51
Immediately after the last treatment	69.50 *	9.07	85.23 *	3.61	88.67 *	5.24
Δ	6.00	3.43	9.55 ^†^	2.13	16.33 ^§,‡^	5.50

Group 1, patients who did not undergo intensive treatment; group 2, patients who underwent intensive treatment for 1 week; group 3, patients who underwent intensive treatment for 2 weeks. Δ, difference between pre-treatment and post-treatment; SD, standard deviation. * *p* < 0.05 according to paired *t*-test and Wilcoxon signed rank test between baseline and post-treatment in each group. ^†^
*p* < 0.05 according to KW test, post hoc analysis (BFT)—groups 1 vs. 2. ^‡^
*p* < 0.05 according to KW test, post hoc analysis (BFT)—groups 1 vs. 3. ^§^
*p* < 0.05 according to KW test, post hoc analysis (BFT)—groups 2 vs. 3.

**Table 3 children-10-01088-t003:** Comparison of changes in SCM thickness (mm) among the three groups.

Variables	Group 1 (*n* = 10)		Group 2 (*n* = 29)		Group 3 (*n* = 15)	
	Mean	SD	Mean	SD	Mean	SD
Pre-treatment	13.36	2.36	11.87	1.48	12.17	1.50
3 months post-treatment	11.81 *	2.60	8.69 *	2.21	8.56 *	1.60
Δ	−1.55	1.28	−3.18 ^†^	2.10	−3.60 ^‡^	2.21

Group 1, patients who did not undergo intensive treatment; group 2, patients who underwent intensive treatment for 1 week; group 3, patients who underwent intensive treatment for 2 weeks. Δ, difference between pre-treatment and post-treatment; SD, standard deviation. * *p* < 0.05 according to paired *t*-test between baseline and post-treatment in each group. ^†^
*p* < 0.05 according to OW-ANOVA, post hoc analysis (BFT)—groups 1 vs. 2. ^‡^
*p* < 0.05 according to OW-ANOVA, post hoc analysis (BFT)—groups 1 vs. 3.

## Data Availability

All data generated/analysed and used to support the findings of this study are included within the article.

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
