# Peer review of "Efficacy of Intensive Inpatient Therapy in Infants with Congenital Muscular Torticollis Involving the Entire Sternocleidomastoid Muscle"

_children, 2023, doi:10.3390/children10061088_

Round 1

Reviewer 1 Report

Thank you for the opportunity to review the manuscript titled “Efficacy of intensive inpatient therapy in infants with congenital muscular torticollis involving the entire sternocleidomastoid muscle”. Although the findings in this paper have the potential to assist clinicians in guiding children and families affected by congenital muscular torticollis, the manuscript is not ready for publication at this time. Reasons are listed below:

       The sample sizes are small when split into cohorts so it is difficult to draw any generalizations about the efficacy of the study. Additionally, group 1 (n=10) is significantly smaller than groups 2 (n=29) and 3 (n=15). This may impact the results.

       Introduction

       Page 2. The statement “Ultrasound has been used as an evaluation tool because of its accuracy detecting morphological changes or muscular pathology” should be expanded upon. While it is discussed in the methods, I would recommend introducing what ultrasound is being used to evaluate in this study here. 

       Figure 1

        First, there is a typo..”outcome date” should be “outcome data”.

       Why does “outcome data” state n=10? This is a bit confusing and suggests that your data analysis was only based on 10 subjects for each group, which does not seem to be the case based on table 2.

       The figure says “Follow up data loss (n=205)”. What is this describing? The manuscript later mentions on page 3 that “rehabilitation treatment was discontinued if normal PCRROM was confirmed.”. Is this what the figure is referring to? A further explanation of data loss is needed.

       Methods

       Page 3: “Groups 2 and 3 underwent the same rehabilitation program twice per day for 1 week”...What is this describing? Some clarification/context is necessary as later in the paragraph the authors state “These exercises were to be performed 10 times per session, six sessions every day”.

       The overall timeline/schedule for each cohort is not very clear. Figure 1 mentions that groups 2 and 3 had intensive treatment for 1 week and 2 weeks respectively, but a different timeline is mentioned later on.

       The authors mention that the home exercise data was collected via “monthly exercise recording sheets” given to parents. This may not be the most reliable source of data.

       Additionally, who taught the parents how to perform home exercises? How long were they trained on how to do this?

       The authors say that all infants were scheduled for routine follow-up appointments with the physiatrist every 4 weeks. The timeline of this part of the study is slightly confusing. I would clarify how long the home exercise program lasted somewhere in the methods section.

       References

       The formatting of references should be fixed. It appears to be single spaced while the rest of the manuscript is double spaced. 

Author Response

Dear the Editors

First, we appreciate you to give us the opportunity to revise our manuscript. Additionally, we appreciate your kindness to detail our manuscript.

The comments were greatly helpful to improve the contents and to revise the errors in our manuscript.

As you mentioned below, we revised our manuscript.

Point to point answers to reviewers are as follows.

Thank you for your consideration.

Sincerely yours.

Dong Rak Kwon, MD, PhD

Department of Rehabilitation Medicine

Catholic University of Daegu School of Medicine

33 Duryugongwon-ro 17-gil, Nam-Gu, Daegu, Korea, 705-718

Phone: +82 53 650 4687  Fax: +82 53 622 4687 

E-mail: coolkwon@cu.ac.kr

Point by Point Answer to Editor

ID: children-2318497

Authors: Dong Rak Kwon and Sung Cheol Cho

Affiliations:

  1. Department of Rehabilitation Medicine, Catholic University of Daegu School of Medicine, Daegu, South Korea.

Title:    Efficacy of intensive inpatient therapy in infants with congenital muscular torticollis involving the entire sternocleidomastoid muscle

All changes made to the manuscript are colored red. Below is a list of reviewers’ comments (C), and our corresponding responses.(R)

<Reviewer 1 Comment>

C1. The sample sizes are small when split into cohorts so it is difficult to draw any generalizations about the efficacy of the study. Additionally, group 1 (n=10) is significantly smaller than groups 2 (n=29) and 3 (n=15). This may impact the results.
R1:
Thank you for your comment. We strongly agree with your opinion. Our study is a retrospective study that involved chart review, and as only one of the three pediatric physical therapists is responsible for congenital torticollis therapy, patients had to wait for physical therapy. Out of the 1,131 diagnosed, 259 met the inclusion criteria, which was more frequent than the previous study (18) (136 out of 1086). The timing of congenital muscular torticollis treatment in the recovery of muscle function is crucial to the outcome. We recommended hospitalization within one month of birth. Although we recommended two weeks of treatment for all patients, some cases varied based on their parents' preferences. 205 patients could not wait for hospitalization and went to other hospitals, 15 received one week of intensive treatment, and 10 did not want intensive treatment. Finally, we analyzed 54 patients: 29 patients in the 2-week intensive care group, 15 patients in the 1-week intensive care group, and 10 patients in the group who did not receive intensive care in the hospital. We added the above sentences to the result section of our paper. As we also mentioned in the discussion, we consider the small sample size a limitation of our study. Therefore, we agree that further studies with larger sample sizes are necessary to confirm our findings.

Reference 18.: Cheng JC, Tang SP, Chen TM, Wong MW, Wong EM. The clinical presentation and outcome of treatment of congenital muscular torticollis in infants—a study of 1,086 cases. J Pediatr Surg 2000; 35: 1091-1096.

C2. Introduction

Page 2. The statement “Ultrasound has been used as an evaluation tool because of its accuracy detecting morphological changes or muscular pathology” should be expanded upon. While it is discussed in the methods, I would recommend introducing what ultrasound is being used to evaluate in this study here.

R2: Thank you for your comment. We added the following sentence in the introduction section to .

Introduction

Ultrasound has been used as an evaluation tool because of its accuracy in detecting morphological changes or muscular pathology such as myopathy, cerebral palsy etc. (12-16). Furthermore, The B-mode ultrasound of the affected sternocleidomastoid muscle includes the echogenicity, texture, and transverse and longitudinal extents of involvement. The echo texture is always more hyperechoic than that of the healthy contralateral or unaffected adjacent muscle. The hyperechogenicity remains the most striking and reliable feature that is correlated with the severity of CMT. The hyperechogenicity in the muscles of neuromuscular disorders is thought to be mainly caused by the replacement of muscle tissue with fat and fibrosis.(16)

Reference 16. Pillen S, Arts IM, Zwarts MJ. Muscle ultrasound in neuromuscular disorders.Muscle Nerve 2008; 37:679–693.

C3. Figure 1

First, there is a typo..”outcome date” should be “outcome data”

R3: Thank you. We corrected the word.

C4. Why does “outcome data” state n=10? This is a bit confusing and suggests that your data analysis was only based on 10 subjects for each group, which does not seem to be the case based on table 2.

R4: Thank you for your valuable comment. As your comment, we revised figure 2 as below

C5. The figure says “Follow up data loss (n=205)”. What is this describing? The manuscript later mentions on page 3 that “rehabilitation treatment was discontinued if normal PCRROM was confirmed.”. Is this what the figure is referring to? A further explanation of data loss is needed.

R5: Thank you for your valuable comment. As your comment, we revised as below.

Results

Out of the 1,131 diagnosed, 259 met the inclusion criteria. We recommended hospitalization within one month of birth. Although we recommended two weeks of treatment for all patients, some cases varied based on their parents' preferences. 205 patients could not wait for hospitalization and went to other hospitals, 15 received one week of intensive treatment, and 10 did not want intensive treatment. Finally, we analyzed 54 patients: 29 patients in the 2-week intensive care group, 15 patients in the 1-week intensive care group, and 10 patients in the group who did not receive intensive care in the hospital. (Figure 2)

C6. Methods

Page 3: “Groups 2 and 3 underwent the same rehabilitation program twice per day for 1 week”...What is this describing? Some clarification/context is necessary as later in the paragraph the authors state “These exercises were to be performed 10 times per session, six sessions every day”.

R6: Thank you for your helpful comment for the improvement of our manuscript. We added the sentence as follows.

Groups 2 and 3 underwent the same rehabilitation program twice per day for 1 week (total: 10 times) and 2 weeks (total: 20 times) during the hospitalization period, respectively.

C7. The overall timeline/schedule for each cohort is not very clear. Figure 1 mentions that groups 2 and 3 had intensive treatment for 1 week and 2 weeks respectively, but a different timeline is mentioned later on.

R7: Thank you for your valuable comment. We added the follow sentences to clarify timeline.

Groups 2 and 3 underwent the same rehabilitation program twice per day for 1 week (total: 10 times) and 2 weeks (total: 20 times) during the hospitalization period, respectively. After being discharged to home, the parents were instructed and encouraged to follow a home exercise program for the infants, consisting of lateral flexion and rotation of the neck to both sides.

We educated home exercise program at our outpatient clinic during first visit by the physiatrist all parents. The physical therapist taught the home exercise program at the end of each treatment session. These exercises were to be performed 10 times per session, six sessions every day.

C8. The authors mention that the home exercise data was collected via “monthly exercise recording sheets” given to parents. This may not be the most reliable source of data.

R8: We agree with your comment. It would be better to evaluate and educate patients in our clinic more often, but it’s unrealistic. The use of digital recording in real time via mobile application would make the outcome more reliable. We mentioned it on limitation.

Discussion

Fifth, infants needed to be immobile during ultrasound examination due to motion artifacts, but it is sometimes very difficult. Herein, all images were obtained after the infants had fallen asleep. Sixth, home exercise data were collected using monthly exercise recording sheets that had been recorded by the parents. The use of digital recording in real time via mobile application would make the outcome more reliable.

C9. Additionally, who taught the parents how to perform home exercises? How long were they trained on how to do this?

R9: Thank you for your valuable comment. We added the follow sentences to clarify home exercises.

Methods

Groups 2 and 3 underwent the same rehabilitation program twice per day for 1 week (total: 10 times) and 2 weeks (total: 20 times) during the hospitalization period, respectively. After being discharged to home, the parents were instructed and encouraged to follow a home exercise program for the infants, consisting of lateral flexion and rotation of the neck to both sides.

We educated home exercise program at our outpatient clinic during first visit by the physiatrist all parents. The physical therapist taught the home exercise program at the end of each treatment session.

C10. The authors say that all infants were scheduled for routine follow-up appointments with the physiatrist every 4 weeks. The timeline of this part of the study is slightly confusing. I would clarify how long the home exercise program lasted somewhere in the methods section.

R: Thank you for your valuable comment. We revised it more clearly.

Methods

The rehabilitation treatment was discontinued if normal PCRROM was confirmed. The home exercise program was recommended to be continued until the discontinuation of the treatment, and we encouraged parents to continue the program after the treatment period. The duration of the home exercise program was not predetermined, and it varied among patients depending on their individual needs.

Limitation section. Seventh, we did not consider the influence of home exercise program duration. The efficacy of the home exercise program beyond the treatment period needs to be investigated in future studies.

C11. References

The formatting of references should be fixed. It appears to be single spaced while the rest of the manuscript is double spaced.

R11: Thank you for your valuable comment. We revised it double spaced.

Reviewer 2 Report

I am grateful for their work in the development of an effective and intensive hospital therapy for infants with congenital muscular torticollis. However, several modifications are necessary to improve its value and suitability for publication. Please address each of the proposed modifications.

The total size of patients constituting the final sample and the patient demographic data should appear in the results section, not in the methods section.

The results section should be improved by including the data of the patients who finally constituted the sample. It is also recommended to separate the data provided in a single table into several tables in order to be able to analyze them more clearly.

Of a total of 20 bibliographic references, four (11,12,13,14) are self-references. Reference 11 justifies its suitability in the explanation of the text but references 12,13 and 14 appear as a set to justify the use of ultrasound. It is recommended to modify them for more recent ones and, if not, to specifically justify each reference in relation to the text.

Author Response

Dear the Editors

First, we appreciate you to give us the opportunity to revise our manuscript. Additionally, we appreciate your kindness to detail our manuscript.

The comments were greatly helpful to improve the contents and to revise the errors in our manuscript.

As you mentioned below, we revised our manuscript.

Point to point answers to reviewers are as follows.

Thank you for your consideration.

Sincerely yours.

Dong Rak Kwon, MD, PhD

Department of Rehabilitation Medicine

Catholic University of Daegu School of Medicine

33 Duryugongwon-ro 17-gil, Nam-Gu, Daegu, Korea, 705-718

Phone: +82 53 650 4687  Fax: +82 53 622 4687 

E-mail: coolkwon@cu.ac.kr

Point by Point Answer to Editor

ID: children-2318497

Authors: Dong Rak Kwon and Sung Cheol Cho

Affiliations:

  1. Department of Rehabilitation Medicine, Catholic University of Daegu School of Medicine, Daegu, South Korea.

Title:    Efficacy of intensive inpatient therapy in infants with congenital muscular torticollis involving the entire sternocleidomastoid muscle

All changes made to the manuscript are colored red. Below is a list of reviewers’ comments (C), and our corresponding responses.(R)

<Reviewer 2 Comment>

C1. The total size of patients constituting the final sample and the patient demographic data should appear in the results section, not in the methods section.
R1: Thank you for your valuable comment. We revised as below.

Methods

This study retrospectively identified 1131 potential participants from a chart review. Of them, 54 were included as research participants. Clinical information was assessed and evaluated. Figure 1 shows an overview of the study protocol. The study was approved by the Institutional Review Board (IRB) and the Independent Ethics Committee of the University Medical Center (IRB No. CR-22-013). Patient consent was waived due to the retrospective nature of this study.

Infants’ age, sex, weight, and height did not significantly differ among the three groups (Table 1).

Results

Out of the 1,131 diagnosed, 259 met the inclusion criteria. We recommended hospitalization within one month of birth. Although we recommended two weeks of treatment for all patients, some cases varied based on their parents' preferences. 205 patients could not wait for hospitalization and went to other hospitals, 15 received one week of intensive treatment, and 10 did not want intensive treatment. Finally, we analyzed 54 patients: 29 patients in the 2-week intensive care group, 15 patients in the 1-week intensive care group, and 10 patients in the group who did not receive intensive care in the hospital. (Figure 2)

Figure 2. Flow chart of the study.

Demographic data at baseline among the three groups were not significantly different (Table 1).

Table 1. Patient demographic data.

Group 1 (n = 10)

Group 2 (n = 29)

Group 3(n = 15)

Corrected age, days

19.60 ± 5.52

20.40 ± 9.01

18.34 ± 15.22

Sex, Male

6 (60%)

13 (45%)

8 (53%)

Affected side, right

6 (60%)

15 (52%)

7 (47%)

Cervical rotation passive range of motion (°)

63.50 ± 10.53

76.21 ± 5.11

72.33 ± 6.51

SCM thickness (mm)

13.36 ± 2.36

11.87 ± 1.48

12.17 ± 1.50

Values are presented as mean ± standard deviation or number (%).

Group 1, patients who did not undergo intensive treatment; Group 2, patients who underwent intensive treatment for 1 week; Group 3, patients who underwent intensive treatment for 2 weeks.

*p < 0.05 by chi-square test.

C2. The results section should be improved by including the data of the patients who finally constituted the sample. It is also recommended to separate the data provided in a single table into several tables in order to be able to analyze them more clearly.

R: Thank you for your valuable comment. We separated the table 2 as below.

Table 2. Comparison of changes of passive cervical rotational range (°) and SCM thickness among the three groups.

Group 1 (n = 10)

Group 2 (n = 29)

Group 3(n = 15)

Pre-treatment

63.50 ± 10.53

76.21 ± 5.11

72.33 ± 6.51

Post-treatment

69.50 ± 9.07*

85.23 ± 3.61*

88.67 ± 5.24*

Δ

6.00 ± 3.43

9.55 ± 2.13

16.33 ± 5.50‡§

Values are presented as mean ± standard deviation. Group 1, patients who did not undergo intensive treatment; Group 2, patients who underwent intensive treatment for 1 week; Group 3, patients who underwent intensive treatment for 2 weeks. Δ, difference between pre-treatment and post-treatment.

*) p < 0.05 by Paired-T test between baseline and post-treatment in each group.

†) p < 0.05 one-way ANOVA, Bon ferroni’s post hoc test between groups 1 and 2.

‡) p < 0.05 one-way ANOVA, Bon ferroni’s post hoc test between groups 1 and 3.

  • ) p < 0.05 one-way ANOVA, Bon ferroni’s post hoc test between groups 2 and 3.

Table 3. Comparison of changes of SCM thickness (mm) among the three groups.

Variables

Group 1 (n = 10)

Group 2 (n = 29)

Group 3(n = 15)

Pre-treatment

13.36 ± 2.36

11.87 ± 1.48

12.17 ± 1.50

3-month post-treatment

11.81 ± 2.60*

8.69 ± 2.21*

8.56 ± 1.60*

Δ

-1.55 ± 1.28

-3.18 ± 2.10

-3.60 ± 2.21

Values are presented as mean ± standard deviation. Group 1, patients who did not undergo intensive treatment; Group 2, patients who underwent intensive treatment for 1 week; Group 3, patients who underwent intensive treatment for 2 weeks. Δ, difference between pre-treatment and post-treatment.

*) p < 0.05 by Paired-T test between baseline and post-treatment in each group.

†) p < 0.05 one-way ANOVA, Bon ferroni’s post hoc test between groups 1 and 2.

‡) p < 0.05 one-way ANOVA, Bon ferroni’s post hoc test between groups 1 and 3.

  • ) p < 0.05 one-way ANOVA, Bon ferroni’s post hoc test between groups 2 and 3.

C3. Of a total of 20 bibliographic references, four (11,12,13,14) are self-references. Reference 11 justifies its suitability in the explanation of the text but references 12,13 and 14 appear as a set to justify the use of ultrasound. It is recommended to modify them for more recent ones and, if not, to specifically justify each reference in relation to the text.

R3: Thank you for your valuable comment. We modified references for more recent ones and added as follows in the revised text.  

Ultrasound has been used as an evaluation tool because of its accuracy in detecting morphological changes or muscular pathology such as myopathy, cerebral palsy etc. (12-16)

  1. Han MH, Kang JY, Do HJ, Park HS, Noh HJ, Cho YH, Jang DH. Comparison of Clinical Findings of Congenital Muscular Torticollis Between Patients With and Without Sternocleidomastoid Lesions as Determined by Ultrasonography. J Pediatr Orthop 2019; 39: 226-231.
  2. Kwon DR, Kim Y. Sternocleidomastoid size and upper trapezius muscle thickness in congenital torticollis patients: A retrospective observational study. Medicine (Baltimore) 2021; 100: e28466.
  3. Kwon DR, Kwon DG. Botulinum Toxin a Injection Combined with Radial Extracorporeal Shock Wave Therapy in Children with Spastic Cerebral Palsy: Shear Wave Sonoelastographic Findings in the Medial Gastrocnemius Muscle, Preliminary Study. Children (Basel) 2021; 8: 1059.
  4. Park GY, Kwon DR, Kwon DG. Shear wave sonoelastography in infants with congenital muscular torticollis. Medicine (Baltimore) 2018; 97: e9818.
  5. Pillen S, Arts IM, Zwarts MJ. Muscle ultrasound in neuromuscular disorders. Muscle Nerve 2008; 37: 679–693.

Reviewer 3 Report

This is an interesting potential addition to the congenital muscular torticollis literature given the following clarifications.

1.  What was the rationale for the duration of stretch being held only for 1 second?

2.  The author should explain the theoretical basis of use of (a) microcurrent therapy and (b) ultrasound diathermy?  What are the potential contraindications for use of each in infants and potential structural growth (if any)?

The changes in the structural composition of the SCM are compelling but the authors need to explain the use of cervical rotation passive range of motion without consideration of lateral flexion motion or any functional, active motion outcomes.  Are the passive motion results felt to be sufficient to support discontinuation of therapy?

Author Response

Dear the Editors

First, we appreciate you to give us the opportunity to revise our manuscript. Additionally, we appreciate your kindness to detail our manuscript.

The comments were greatly helpful to improve the contents and to revise the errors in our manuscript.

As you mentioned below, we revised our manuscript.

Point to point answers to reviewers are as follows.

Thank you for your consideration.

Sincerely yours.

Dong Rak Kwon, MD, PhD

Department of Rehabilitation Medicine

Catholic University of Daegu School of Medicine

33 Duryugongwon-ro 17-gil, Nam-Gu, Daegu, Korea, 705-718

Phone: +82 53 650 4687  Fax: +82 53 622 4687 

E-mail: coolkwon@cu.ac.kr

Point by Point Answer to Editor

ID: children-2318497

Authors: Dong Rak Kwon and Sung Cheol Cho

Title:    Efficacy of intensive inpatient therapy in infants with congenital muscular torticollis involving the entire sternocleidomastoid muscle

All changes made to the manuscript are colored red. Below is a list of reviewers’ comments (C), and our corresponding responses.(R)

Reviewer 3 comment

C1. What was the rationale for the duration of stretch being held only for 1 second?

R1: Thank you for your valuable comment. The duration of the stretch being held was 10 seconds. It was a typographical error in our manuscript, and we have made the necessary correction.

C2. The author should explain the theoretical basis of use of (a) microcurrent therapy and (b) ultrasound diathermy?  What are the potential contraindications for use of each in infants and potential structural growth (if any)?

R2: Thank you for your valuable comment. In our revised manuscript, we have incorporated the following sentence into the introduction section to provide the theoretical background of both microcurrent therapy and ultrasound diathermy. Additionally, it is

important to note that the infants included in the study had no contraindications.

MCT has been found to have therapeutic effects on muscle damage, attributed to various mechanisms. Firstly, it is believed that microcurrent therapy helps maintain intracellular calcium (Ca2+) balance following muscle damage, as demonstrated in a previous study (14). By regulating intracellular calcium levels, this therapy can potentially prevent disruptions to the muscle membrane integrity and detrimental changes in muscle function caused by increased calcium concentration (15). In the context of CMT, where sustained isometric contraction of the SCM can impede regular blood flow and ion exchange at the cellular membrane, this mechanism becomes particularly relevant (16, 17).

Secondly, MCT has the potential to improve ATP production, facilitate amino acid transportation, and promote protein production. These processes play crucial roles in reducing inflammation and promoting tissue healing (18, 19). Previous studies have also highlighted the beneficial effects of electric current in repairing soft tissues (20).

Lastly, MCT may influence the cell nuclei activity and triggers the genes responsible for regulating collagen depletion and breaking, thus alleviating fibrosis. Notably, in patients with head and neck carcinoma having radiation-induced fibrosis, the initiation of microcurrent therapy resulted in a significant improvement in cervical rotational range of motion (18).

Previous studies have indeed supported the efficacy of therapeutic US in reducing inflammation and enhancing blood circulation in treated tissues, given that the application is conducted appropriately (21, 22). Furthermore, a recent animal study provided additional evidence of the benefits of therapeutic US. The study demonstrated that applying therapeutic US was efficient in guarding the muscle fibers from getting into oxidative stress, mitigating the inflammatory reactions, and promoting the reorganization of muscle tissues, as observed in immunohistochemical analysis (23).

C3. The changes in the structural composition of the SCM are compelling but the authors need to explain the use of cervical rotation passive range of motion without consideration of lateral flexion motion or any functional, active motion outcomes.  Are the passive motion results felt to be sufficient to support discontinuation of therapy?

R3: Thank you for your comments. In our revised manuscript, we have included specific criteria for discontinuing treatment to enhance clarity and comprehensibility.

All infants were scheduled for routine follow-up examinations with the physiatrist every 4 weeks. In addition, the infants were rescheduled for evaluation by the physiatrist within 1 week if the physical therapist detected a normal passive cervical rotational range of motion (PCRROM) on the affected side during the treatment session. The rehabilitation treatment was discontinued if patients met following 5 criteria: (1) difference of cervical passive range of motion is within 5° of the nonaffected side; (2) symmetrical movement patterns are observed; (3) proper motor development with regard to age is achieved; (4) no head tilt during active or static postures is observed; and (5) the parents clearly understand what to monitor as the child grows (28).

Reviewer 4 Report

Overall, this study provides valuable insights into the short-term effects of intensive inpatient rehabilitation programs for infants with CMT involving the entire SCM muscle. The findings have implications for improving treatment outcomes and highlight the potential benefits of the described therapy, optimizing treatment sessions. However, the authors need to provide more detailed information on what is already known about the topic, how the intervention programs were planned, and edit the manuscript to reflect the way manuscripts should look like (e.g., the usage of customary sub-titles).  

Abstract

·         Write a description of treatment 1 before 2 and 3

·         Provide basic demographic details of participants (e.g., sex)

Introduction

·         In all references, the period should be after the reference and not before.

·         In the first paragraph add information on CMT epidemiology

·         Active or passive stretching of the affected muscle may be effective only in infants in whom less than two-thirds of the SCM muscle is involved.(9)  - what about the other third?

·         You didn't provide enough information on the ultrasound therapy for this condition – please elaborate more. What is known/not known about this treatment?

·         Please add information on the mechanisms of change following ultrasound. What are its special benefits for the studied condition?

·          Add study hypothesis

Methods

·         Before figures please write a description of the figure and only then show the figure.

·         Add title participants and in it write about the inclusion and exclusion criteria

·         Add title procedures and in it describe the groups and what each group received

·         Add a section of outcome measures

·         Add title statistical analysis

·         Did you do a power analysis?

·         Did you check for data normality?

·         Please provide more rationale for the treatment protocol. Why 1 week? Why 2 weeks? What is typically conducted today?

Results

·         Add a section of participants' demographic characteristics

·         Table 1 – please add a column with the F and p values.

·         Table 1 – please add measurement values to all variables

·         Table 1 – cervical rotation, SCM thickness are not demographic data. Please write age and sex in the text and not in the table. The other variables in Table 1 are not needed. They also appear in the other tables.

·         Tables 2 and 3– add the values of the number mean and SD in the first row and not in the notes. Look in other manuscripts to see how the units are described in tables. 

·         Table 2 says post-treatment and Table 3 – 3 months post-treatment. Is in Table 2 it is also 3 months? Please have consistent names in both tables.

·         I think it will also be useful to calculate effect sizes for each study group. 

Discussion

·         The differences observed among the groups can be attributed to several factors, such as the management of the home program, and the inclusion of microcurrent therapy. Please, elaborate on the influence of confounding variables. 

Author Response

Dear the Editors

First, we appreciate you to give us the opportunity to revise our manuscript. Additionally, we appreciate your kindness to detail our manuscript.

The comments were greatly helpful to improve the contents and to revise the errors in our manuscript.

As you mentioned below, we revised our manuscript.

Point to point answers to reviewers are as follows.

Thank you for your consideration.

Sincerely yours.

Dong Rak Kwon, MD, PhD

Department of Rehabilitation Medicine

Catholic University of Daegu School of Medicine

33 Duryugongwon-ro 17-gil, Nam-Gu, Daegu, Korea, 705-718

Phone: +82 53 650 4687  Fax: +82 53 622 4687 

E-mail: coolkwon@cu.ac.kr

Point by Point Answer to Editor

ID: children-2318497

Authors: Dong Rak Kwon and Sung Cheol Cho

Title:    Efficacy of intensive inpatient therapy in infants with congenital muscular torticollis involving the entire sternocleidomastoid muscle

All changes made to the manuscript are colored red. Below is a list of reviewers’ comments (C), and our corresponding responses.(R)

Reviewer 4 comment

C1. Abstract

-Write a description of treatment 1 before 2 and 3

 -  Provide basic demographic details of participants (e.g., sex)

R1 : Thank you for your comments. We revised as follows according to your comment.

This study included 54 infants (27 boys and 27 girls; mean corrected age of 18.57 days) evaluated for CMT at our outpatient clinic from January 2014 to May 2021. The included patients were divided into three groups (groups 1, 2, and 3). Patients in group 1 underwent outpatient treatment for 12 times. Patients in groups 2 and 3 underwent therapeutic exercise followed by US diathermy with microcurrent twice daily for 1 and 2 weeks, respectively.

C2.  Introduction

- In all references, the period should be after the reference and not before.

  1. We did it.

 - In the first paragraph add information on CMT epidemiology

R : We added information on CMT epidemiology in the first paragraph.

Congenital muscular torticollis (CMT) is the 3rd common congenital anomaly among infants, next only to congenital hip dysplasia and clubfoot. The reported incidence of CMT varies between 0.4% and 2.0% (1).

- Active or passive stretching of the affected muscle may be effective only in infants in whom less than two-thirds of the SCM muscle is involved.(9)  - what about the other third?

R : In case of entire fibrosis of SCM, it is estimated that approximately one-third of patients cannot be effectively treated with physical therapy alone. Consequently, more invasive interventions like surgery and botulinum neurotoxin injections may be necessary to treat the contracture (11).

- You didn't provide enough information on the ultrasound therapy for this condition – please elaborate more. What is known/not known about this treatment?

- Please add information on the mechanisms of change following ultrasound. What are its special benefits for the studied condition?

R : In our revised manuscript, we have incorporated the following sentence into the introduction section to provide the theoretical background of ultrasound diathermy.

Previous studies have indeed supported the efficacy of therapeutic US in reducing inflammation and enhancing blood circulation in treated tissues, given that the application is conducted appropriately (21, 22). Furthermore, a recent animal study provided additional evidence of the benefits of therapeutic US. The study demonstrated that applying therapeutic US was efficient in guarding the muscle fibers from getting into oxidative stress, mitigating the inflammatory reactions, and promoting the reorganization of muscle tissues, as observed in immunohistochemical analysis (23).

- Add study hypothesis

R : We added the study hypothesis.

Our hypothesis suggests that a two-week inpatient program, characterized by intensive treatment, is expected to yield superior treatment outcomes compared to a one-week intensive treatment program or an outpatient-based treatment approach.

C3 Method

- Before figures please write a description of the figure and only then show the figure.

R : We revised as your comment

- Add title participants and in it write about the inclusion and exclusion criteria

- Add title procedures and in it describe the groups and what each group received

- Add a section of outcome measures

- Add title statistical analysis

R : We have revised the manuscript according to your comments above.

- Did you do a power analysis?

R : Thank you for your comment. In our study, we did not conduct a sample size calculation due to the absence of previous studies similar to ours. Instead, we performed a retrospective power analysis to estimate the statistical power of our study. The power for cervical range of motion (ROM) was found to be above 0.8, indicating a sufficient sample size for that particular outcome. However, we observed that the power for SCM thickness was 0.68, suggesting that a larger sample size would be necessary to obtain more precise results for this variable. In future studies, we plan to address this limitation by recruiting a larger number of infants to improve the accuracy of our findings. We added the limitation as follow.

Discussion

Tenth, we did not conduct a sample size calculation due to the absence of previous studies similar to ours we performed a retrospective power analysis to estimate the statistical power of our study. The power for cervical range of motion (ROM) was found to be above 0.8, indicating a sufficient sample size for that particular outcome. However, we observed that the power for SCM thickness was 0.68, suggesting that a larger sample size would be necessary to obtain more precise results for this variable. In future studies, we plan to address this limitation by recruiting a larger number of infants to improve the accuracy of our findings.

- Did you check for data normality?

R : Yes we did it. The data in each group showed normality therefore we did parametric method, independent t-test and ANOVA.

Please provide more rationale for the treatment protocol. Why 1 week? Why 2 weeks? What is typically conducted today?

R : Thank you for your valuable comment. We added following sentence in discussion section.

In our study, we proposed a standardized treatment duration of two weeks for all patients. It is important to note that the optimal length of inpatient treatment for this particular condition is not well-established. Our observations indicate that there were difficulties in sustaining hospitalization for longer than two weeks. Consequently, the actual duration of hospitalization varied among patients, taking into consideration the preferences of parents or caregivers involved in the decision-making process.

C4 Result

- Add a section of participants' demographic characteristics

-Table 1 – please add a column with the F and p values.

-Table 1 – please add measurement values to all variables

-Table 1 – cervical rotation, SCM thickness are not demographic data. Please write age and sex in the text and not in the table. The other variables in Table 1 are not needed. They also appear in the other tables.

R : In the revised manuscript, we have removed Table 1 in response to your comment and rewritten the text as a description of the results. Instead, we have created a Table 1 that compares the outcomes of the inpatient intensive and outpatient groups, which is an important outcome.

Group 1 had 10 infants (6 boys and 4 girls; affected side, 6 right, 4 left; mean corrected age, 19.60 ± 5.52 days), group 2 had 29 infants (13 boys and 16 girls; affected side, 15 right, 14 left; mean corrected age, 20.40 ± 9.01 days), and group 3 had 15 infants (8 boys and 7 girls; affected side, 7 right, 8 left; mean corrected age, 18.34 ± 15.22 days). There were no statistically significant differences in corrected age (F = 0.412, p-value = 0.873), sex (F = 0.698, p-value = 0.794), affected side (F = 0.817, p-value = 0.618) between the three groups. Demographic data at baseline among the three groups were not significantly different.

In the inpatient treatment groups (group 2 and 3), both the thickness of SCM and the PCRROM showed a significant decrease/increase when compared to the outpatient treatment group (group 1) (p < 0.05, Table 1).

All three groups had a significant decrease in SCM thickness following the respective group intervention (p < 0.05, Table 2). Additionally, all three groups had a significant increase in PCRROM after treatment compared to before treatment (p < 0.05, Table 3). The greatest mean change in PCRROM was observed in group 3 (16.33° ± 5.50°), followed by group 2 (9.55° ± 2.13°) and group 1 (6.00° ± 3.43°) (p < 0.05, Table 2). Furthermore, the mean reduction in SCM thickness between the pre- and 3-month post-treatment periods was significantly greater in Group 2 (-3.18 mm ± 2.10 mm) and Group 3 (-3.60 mm ± 2.21 mm) than in Group 1 (-1.55 mm ± 1.28 mm) (p < 0.05, Table 3).

Table 1. Comparison of changes in the passive cervical rotational range (°) and SCM thickness between outpatient and inpatient treatment groups.

Outpatient treatment (n = 10)

Inpatient treatment (n = 44)

Mean

SD

Mean

SD

Cervical rotational PROM (°)

Pre-treatment

63.50

10.53

74.89

5.86

Immediately after the last treatment

69.50 

9.07

86.59

3.70

Δ

6.00

3.43

11.70*

4.94

SCM thickness (mm)

Pre-treatment

13.36

2.36

11.97

1.48

3-month post-treatment

11.81

2.60

9.44

2.07

Δ

−1.55

1.28

−3.58*

1.85

Outpatient treatment, patients who did not undergo intensive treatment; Inpatient treatment, patients who undergo intensive treatment. Δ, difference between pre-treatment and post-treatment.

Δ, difference between pre-treatment and post-treatment.

*) p < 0.05 by independent T test between inpatient and outpatient treatment patients.

-Tables 2 and 3– add the values of the number mean and SD in the first row and not in the notes. Look in other manuscripts to see how the units are described in tables. 

R: In revised manuscript, we added it as your comments.

Table 2 says post-treatment and Table 3 – 3 months post-treatment. Is in Table 2 it is also 3 months? Please have consistent names in both tables.

R : Thank you for your comments. In our revised manuscript, we have included specific value names to enhance clarity and comprehensibility.

I think it will also be useful to calculate effect size for each study group.

R : Thank you for your comment. We appreciate your suggestion to calculate the effect size for each study group. However, in our study, we conducted a retrospective power analysis and did not calculate the sample size. As a result, we were unable to calculate the effect size for each group. We acknowledge that calculating effect sizes would provide valuable information and could be considered in future research endeavors.

C5 Discussion

 The differences observed among the groups can be attributed to several factors, such as the management of the home program, and the inclusion of microcurrent therapy. Please, elaborate on the influence of confounding variables. 

R5: Thank you for your advice. We explained as limitation in discussion section as follows.

Sixth, home exercise databases were gathered through monthly exercise recording sheets that were completed by the parents. The use of digital recording in real time via mobile applications would make the outcome more reliable. Seventh, we did not consider the influence of home exercise program duration. The efficacy of the home exercise program beyond the treatment period needs to be investigated in future studies. Eighth, we only evaluated the PCRROM as the outcome measurement. Evaluation of cervical lateral flexion range of motion would have made the result more reliable.

Ninth, further research is necessary to assess the effects of MCT using different current intensities, waveforms, frequencies, and treatment durations. This research will help identify the optimal parameters that yield the most effective and consistent results. Tenth, we did not conduct a sample size calculation due to the absence of previous studies similar to ours we performed a retrospective power analysis to estimate the statistical power of our study. The power for cervical range of motion (ROM) was found to be above 0.8, indicating a sufficient sample size for that particular outcome. However, we observed that the power for SCM thickness was 0.68, suggesting that a larger sample size would be necessary to obtain more precise results for this variable. In future studies, we plan to address this limitation by recruiting a larger number of infants to improve the accuracy of our findings.

Finally, infants whose treatment started after 1 month of age were not included in this study. The prognosis for CMT generally based on the age at which the rehabilitation program was started. Further researches are necessary to analyse the effects of rehabilitation programs according to the age at which the treatment is initiated in infants with CMT including the entire SCM muscle.

Round 2

Reviewer 1 Report

Thank you for responding to our comments on your manuscript titled “Efficacy of intensive inpatient therapy in infants with congenital muscular torticollis involving the entire sternocleidomastoid muscle”. There are still many changes that would need to be made in order for this manuscript to be ready for publication.

       Generally, the entire structure of the paper needs to be moved around. The methods section should discuss why so many subjects were included. Figure 1 fits better in the methods than in the results.

Introduction

       Page 1: The sentence “One-third of cases where fibrosis affects the entire SCM muscle length cannot be resolved by physical therapy and require surgery to release the contracted muscles.” should be reworded. The way it currently reads makes it sound as though one-third of cases affect the entire SCM muscle length, and not that one-third of cases cannot be resolved by physical therapy.

       The sentence “The physical therapist performed the same rehabilitation program which consisted of therapeutic exercise with ultrasound and microcurrent therapy in three groups” is confusing and may need to be moved. The word “same” does not make sense here because no rehabilitation program has been mentioned prior to this sentence.

Methods:

       Figure 1 should be placed back in the beginning of the methods, along with the explanation for why many patients did not meet criteria.

       Page 3: The sentence “We educated home exercise program at our outpatient clinic during first visit by the physiatrist all parents. The physical therapist taught the home exercise program at the end of each treatment session.” is grammatically incorrect. It should be rephrased.

       The sentence “The physical therapist performed the same rehabilitation program which consisted of therapeutic exercise with ultrasound and microcurrent therapy in three groups” is confusing and may need to be moved. The word “same” does not make sense here because no rehabilitation program has been mentioned prior this sentence.

Table 1:

       The description says, “The mean change of PCRROM was significantly the greatest in group 3 (16.33° ± 5.50°)”. The word “significantly” is unnecessary and should be removed.

Discussion:

       Page 7: They say in limitations age was not contemplated, but state that all infants were younger than 1 month previously. If they are referring to data analysis, the word “contemplated” should be changed. It doesn’t fit right.

Author Response

Dear Editors:

Thank you for giving us the opportunity to revise our manuscript. We appreciate your detailed review of our manuscript.

The comments provided by you and the reviewers have greatly helped us in improving the content of our manuscript and correcting errors.

We have revised our manuscript according to all comments provided.

To improve the text quality and make our arguments clearer to the reader, we had the paper thoroughly proofread by an English native speaker. We were able to improve any awkward expressions and correct grammatical errors.

Please find below our point-by-point responses to the reviewers’ comments.

Thank you once again for your consideration.

Sincerely yours,

Dong Rak Kwon, MD, PhD

Department of Rehabilitation Medicine

Catholic University of Daegu School of Medicine

33 Duryugongwon-ro 17-gil, Nam-Gu, Daegu, Korea, 705-718

Phone: +82 53 650 4687; Fax: +82 53 622 4687

E-mail: coolkwon@cu.ac.kr

Point-by-point responses to reviewers’ comments

ID: children-2318497

Authors: Dong Rak Kwon and Sung Cheol Cho

Affiliation:

  1. Department of Rehabilitation Medicine, Catholic University of Daegu School of Medicine, Daegu, South Korea.

Title:    Efficacy of intensive inpatient therapy in infants with congenital muscular torticollis involving the entire sternocleidomastoid muscle

All changes made to the manuscript are marked in red. Below is a list of reviewers’ comments (C) and our corresponding responses (R).

<Reviewer 1 Comment>

C1. Generally, the entire structure of the paper needs to be moved around. The methods section should discuss why so many subjects were included. Figure 1 fits better in the methods than in the results.

R1:
Thank you for your insightful comment. We have revised the Methods and Results sections of the manuscript accordingly and added details of the exclusion data in Figure 1. Moreover, Figure 1 has been moved to the Methods section as per your suggestion.

Methods

This retrospective chart review identified 1131 potential participants who visited our outpatient clinic with symptoms of torticollis from January 2014 to May 2021. The study was approved by the Institutional Review Board (IRB) and the Independent Ethics Committee of the University Medical Center (IRB No. CR-22-013). Patient consent was waived due to the retrospective nature of this study.

This study included infants with CMT who visited the outpatient clinic of the university hospital and met the following criteria: (1) entire SCM muscle involvement; (2) an SCM muscle thickness of >10 mm, as measured via ultrasound; (3) a palpable mass in the SCM muscle upon clinical examination; (4) age under 3 months; and (5) no previous rehabilitation therapy received before study participation. The exclusion criteria were as follows: patients with (1) postural torticollis, (2) congenital anomalies of the cervical spine, (3) spasmodic torticollis, (4) neurodevelopmental disorders, such as cerebral palsy and intellectual disability, and (5) ocular torticollis.

Results

Out of the 1,131 patients with torticollis, 1077 were excluded. According to the exclusion criteria, 837 infants with postural torticollis, 29 with neurodevelopmental disorders, 3 with congenital anomalies of the cervical spine, 2 with ocular torticollis, and 1 with spasmodic torticollis were excluded. Finally, a total of 259 patients met the inclusion criteria. We recommended hospitalization within 1 month of birth. Although we recommended a standard treatment period of 2 weeks for all patients, the duration varied in some cases based on their parents’ preferences. A total of 205 patients could not wait for hospitalization and went to other hospitals, 15 received 1 week of intensive treatment, and 10 did not want intensive treatment. Finally, we analyzed 54 patients: 29 patients in the 2-week intensive care group, 15 patients in the 1-week intensive care group, and 10 patients in the group who did not receive intensive care in the hospital (Figure 1).

C2. Introduction

Page 1C1: The sentence “One-third of cases where fibrosis affects the entire SCM muscle length cannot be resolved by physical therapy and require surgery to release the contracted muscles.” should be reworded. The way it currently reads makes it sound as though one-third of cases affect the entire SCM muscle length, and not that one-third of cases cannot be resolved by physical therapy.

R2: Thank you for your insightful comment. We have revised the text as follows.

Introduction

In cases where fibrosis affects the entire SCM muscle layer, approximately one-third of the patients cannot be successfully treated with physical therapy alone and may require surgery to release the contracted muscles.

C3. The sentence “The physical therapist performed the same rehabilitation program which consisted of therapeutic exercise with ultrasound and microcurrent therapy in three groups” is confusing and may need to be moved. The word “same” does not make sense here because no rehabilitation program has been mentioned prior to this sentence.

R3: Thank you for your valuable comment. Accordingly, we have deleted the indicated sentence from our article to avoid confusion.

C4. Method

Figure 1 should be placed back in the beginning of the methods, along with the explanation for why many patients did not meet criteria.

R4: Thank you for your valuable comment. Accordingly, we have revised the manuscript and added details regarding the exclusion data in Figure 1. Moreover, Figure 1 has been moved to the beginning of the Methods section.

Out of the 1,131 patients with torticollis, 1077 were excluded. According to the exclusion criteria, 837 infants with postural torticollis, 29 with neurodevelopmental disorders, 3 with congenital anomalies of the cervical spine, 2 with ocular torticollis, and 1 with spasmodic torticollis were excluded. A total of 259 patients met the inclusion criteria. We recommended hospitalization within 1 month of birth. Although we recommended a standard treatment period of 2 weeks for all patients, the duration varied in some cases based on their parents’ preferences. A total of 205 patients could not wait for hospitalization and went to other hospitals, 15 received 1 week of intensive treatment, and 10 did not want intensive treatment. Finally, we analyzed 54 patients: 29 patients in the 2-week intensive care group, 15 patients in the 1-week intensive care group, and 10 patients in the group who did not receive intensive care in the hospital (Figure 1).

C5. Page 3: The sentence “We educated home exercise program at our outpatient clinic during first visit by the physiatrist all parents. The physical therapist taught the home exercise program at the end of each treatment session.” is grammatically incorrect. It should be rephrased.

R5: Thank you for your comments. We have revised it as follows.

During the first visit to our outpatient clinic, the physiatrist educated all parents about the home exercise program, and the physical therapist provided further instructions on the program at the end of each treatment session.

C6. Result.

The description says, “The mean change of PCRROM was significantly the greatest in group 3 (16.33° ± 5.50°)”. The word “significantly” is unnecessary and should be removed.

R6: Thank you for your comments. We have revised it accordingly.

C7. Discussion

Page 7: They say in limitations age was not contemplated, but state that all infants were younger than 1 month previously. If they are referring to data analysis, the word “contemplated” should be changed. It doesn’t fit right.

R7: Thank you for your comments. We have revised it as follows.

Finally, infants whose treatment began after 1 month of age were not included in this study. The prognosis of CMT generally depends on the age at which the rehabilitation program is initiated. Further studies are required to evaluate the effects of rehabilitation programs according to the age at which the therapy is initiated in infants with CMT involving the entire SCM muscle.

Reviewer 2 Report

The paper can be accepted without any further changes.

Author Response

Dear Editors:

Thank you for giving us the opportunity to revise our manuscript. We appreciate your detailed review of our manuscript.

The comments provided by you and the reviewers have greatly helped us in improving the content of our manuscript and correcting errors.

We have revised our manuscript according to all comments provided.

To improve the text quality and make our arguments clearer to the reader, we had the paper thoroughly proofread by an English native speaker. We were able to improve any awkward expressions and correct grammatical errors.

Please find below our point-by-point responses to the reviewers’ comments.

Thank you once again for your consideration.

Sincerely yours,

Dong Rak Kwon, MD, PhD

Department of Rehabilitation Medicine

Catholic University of Daegu School of Medicine

33 Duryugongwon-ro 17-gil, Nam-Gu, Daegu, Korea, 705-718

Phone: +82 53 650 4687; Fax: +82 53 622 4687

E-mail: coolkwon@cu.ac.kr

Reviewer 4 Report

1.       Please add in the data analysis section description of how data normality were evaluated and its results.

2.       Please add more information on how power analysis was computed.  

3.       In the table write in the notes below the table the meaning of SD

4.       Regarding calculating effect size. You can calculate effect size. For example, Cohen's d effect size. It has nothing to do with the fact that you didn't do power analysis. Although it can be used to calculate it.

5.       In limitations correct that you didn't do a priory power analyses. Don’t just write power analysis as you did do a retrospective one.  

Author Response

Dear the Editors

First, we appreciate you to give us the opportunity to revise our manuscript. Additionally, we appreciate your kindness to detail our manuscript.

The comments were greatly helpful to improve the contents and to revise the errors in our manuscript.

As you mentioned below, we revised our manuscript.

Point to point answers to reviewers are as follows.

Thank you for your consideration.

Sincerely yours.

Dong Rak Kwon, MD, PhD

Department of Rehabilitation Medicine

Catholic University of Daegu School of Medicine

33 Duryugongwon-ro 17-gil, Nam-Gu, Daegu, Korea, 705-718

Phone: +82 53 650 4687  Fax: +82 53 622 4687 

E-mail: coolkwon@cu.ac.kr

Point by Point Answer to Editor

ID: children-2318497

Authors: Dong Rak Kwon and Sung Cheol Cho

Title:    Efficacy of intensive inpatient therapy in infants with congenital muscular torticollis involving the entire sternocleidomastoid muscle

Please find enclosed the Certification of medical statistics (pdf) file and the data excel file. The data excel file is not attached, so I have converted it to a pdf file and enclosed it.

All changes made to the manuscript are colored red. Below is a list of reviewers’ comments (C), and our corresponding responses.(R)

C1. Please add in the data analysis section description of how data normality were evaluated and its results.

R1. We sincerely apologize for the previous incorrect response. We have re-evaluated the normality of the data using the appropriate method. The Shapiro-Wilk test was employed to assess normality, and the results are as follows: The data for the normality test has been attached, and you will obtain the same result as us. Notably, the delta of passive cervical rotational range did not demonstrate normality in groups 2 and 3. As a result, the Kruskal-Wallis test, a non-parametric approach, was conducted for comparison. Conversely, the delta of SCM thickness exhibited normality, allowing for analysis using a one-way ANOVA as a parametric approach.

The p-values result of normality test .

Variables

group1

group2

group3

Delta of passive cervical rotational range

0.760

<0.001

0.025

Delta of SCM thickness

0.079

0.382

0.902

C2. Please add more information on how power analysis was computed.  

R2. We sincerely apologize for the previous incorrect response. We have re-evaluated the power of the test using the correct methodology. We used simulation to calculate the power in R program since the patients number in each group were unbalanced. The simulation code was follows and you will get the same result as ours, when you run the below simulation code.

# Number of simulations

n_sim <- 10000

# Sample size of each group

sampsi <- c(10, 29, 15)

# Mean of each group

mus <- c(6,9.55,16.33)

# Standard deviation of each group (assumed to be equal!)

sds <- c(3.43,2.13,5.50)

p_vals <- NULL

# Set seed for reproducibility

set.seed(142857)

for(i in 1:n_sim) {

  dat_tmp <- data.frame(

    y = rnorm(sum(sampsi), mean = rep(mus, times = sampsi), sd = rep(sds, times = sampsi))

    , group = factor(rep(seq_along(mus), times = sampsi))

  )

  mod <- anova(lm(y~group, data = dat_tmp))

  p_vals[i] <-  mod$`Pr(>F)`[1]

  rm(dat_tmp)

}

cat("Simulated power is:", mean(p_vals <= 0.05)*100, "%")

C3. In the table write in the notes below the table the meaning of SD

R3. Thanks for your comments. We note the meaning of SD below the tables.

C4. Regarding calculating effect size. You can calculate effect size. For example, Cohen's d effect size. It has nothing to do with the fact that you didn't do power analysis. Although it can be used to calculate it.

R4. According to the normality test, the delta of passive cervical rotational range was analyzed using the Krsukal-Wallis test, and the delta of SCM thickness was analyzed by one-way ANOVA. Therefore, the effect size was calculated under each analysis method. The effect size was analyzed using the R program and the following R code. The data for effect size calculation was attached and you will get the same result as ours. The effect size of delta of passive cervical rotational range was 0.573, and the effect size of delta of SCM thickness was calculated as 0.09.

install.packages("readxl")

install.packages("effectsize")

install.packages("rstatix")

library(readxl)

library(effectsize)

library(rstatix)

setwd("D:/CMT")

data=read_excel("data.xlsx")

attach(data)

kruskal_effsize(data, Delta_passive_cervical_rotational_range ~ group)

model <- aov(Delta_SCM_thickness ~ group)

eta_squared(model)

C5. In limitations correct that you didn't do a priory power analyses. Don’t just write power analysis as you did do a retrospective one.  

R5. We apologize for the confusion caused by the previous incorrect response. It appears that the previous statement regarding the performance of retrospective power analysis was inaccurate. Upon re-evaluation, the power of the test was correctly performed, and all relevant values have been appropriately corrected in the discussion section of the manuscript.

Discussion

Tenth, we did not do sample size calculation because there was no previous study like our study. We didn’t perform retrospective power analysis. The power of test for cervical ROM was 1.0. But the power for SCM thickness was 0.64. It means more sample size in necessary for exact result. In future study, we will recruit more infant for study.  

Certification of Medical statistics

Data set

Round 3

Reviewer 1 Report

Thank you for the opportunity to review the manuscript titled “Efficacy of intensive inpatient therapy in infants with congenital muscular torticollis involving the entire sternocleidomastoid muscle”. Although the findings in this paper have the potential to assist clinicians in guiding children and families affected by congenital muscular torticollis, the manuscript is not ready for publication at this time and should be revised. Reasons are listed below:

  • Methods: 

    • The sentence describing the timeline is helpful, however, it needs elaboration. Were patients who visited this outpatient clinic with “symptoms of torticollis” diagnosed upon arrival, or diagnosed prior to arrival? This sentence should be reworded to a more matter of fact statement. For example “this study took place using patient visits from __year to ___year”. 

  • Results: 

    • Overall, the results section needs a lot of work. The first sentence should be in the methods, and not the results section. Discussing inclusion/exclusion criteria as well as methodology as a part of the results is not acceptable. 

  • Thank you for changing the introduction sentence about 1/3 of patients. This looks much better.

  • Overall, the paper still needs major restructuring. While there is potential here, large sections need to be moved around.

    • I would ensure that anything in the results section describing includion/exclusion criteria is moved to the methods. Additionally, any references to findings in other papers should be in the discussion or introduction section only. 

Author Response

Dear the Editors

First, we appreciate you to give us the opportunity to revise our manuscript. Additionally, we appreciate your kindness to detail our manuscript.

The comments were greatly helpful to improve the contents and to revise the errors in our manuscript.

As you mentioned below, we revised our manuscript.

Point to point answers to reviewers are as follows.

Thank you for your consideration.

Sincerely yours.

Dong Rak Kwon, MD, PhD

Department of Rehabilitation Medicine

Catholic University of Daegu School of Medicine

33 Duryugongwon-ro 17-gil, Nam-Gu, Daegu, Korea, 705-718

Phone: +82 53 650 4687  Fax: +82 53 622 4687 

E-mail: coolkwon@cu.ac.kr

Point by Point Answer to Editor

ID: children-2318497

Authors: Dong Rak Kwon and Sung Cheol Cho

Title:    Efficacy of intensive inpatient therapy in infants with congenital muscular torticollis involving the entire sternocleidomastoid muscle

All changes made to the manuscript are colored red. Below is a list of reviewers’ comments (C), and our corresponding responses.(R)

<Reviewer 1 Comment>

C1.

Methods

The sentence describing the timeline is helpful, however, it needs elaboration. Were patients who visited this outpatient clinic with “symptoms of torticollis” diagnosed upon arrival, or diagnosed prior to arrival? This sentence should be reworded to a more matter of fact statement. For example “this study took place using patient visits from __year to ___year”

R1:
Thank you for your valuable comment. We revised manuscript as your comment..

This retrospective chart review identified 1131 potential participants who visited our outpatient clinic with symptoms of torticollis from January 2014 to May 2021. The institution conducting the study was a tertiary care center, and all patients included in the study were referred with symptoms of torticollis from other healthcare providers. Referred patients with a palpable mass on physical examination were diagnosed with torticollis involving the entire SCM muscle based on imaging findings such as USG and plain radiographs.

C2.

Result

Overall, the results section needs a lot of work. The first sentence should be in the methods, and not the results section. Discussing inclusion/exclusion criteria as well as methodology as a part of the results is not acceptable.

R2: Thank you for your comment. We revised it as your comment.

C3. Thank you for changing the introduction sentence about 1/3 of patients. This looks much better.

R3: Thank you.

C4. Overall, the paper still needs major restructuring. While there is potential here, large sections need to be moved around.

C5. I would ensure that anything in the results section describing includion/exclusion criteria is moved to the methods. Additionally, any references to findings in other papers should be in the discussion or introduction section only.

R4&5: Thank you for your comment. We have rewritten and revised many parts of the manuscript based on your comments.
